# The *Zmat2* gene in non-mammalian vertebrates: Organizational simplicity within a divergent locus in fish

Peter Rotwein *

Department of Molecular and Translational Medicine, Paul L. Foster School of Medicine, Texas Tech Health University Health Sciences Center, El Paso, Texas, United States of America

* peter.rotwein@ttuhsc.edu

## Abstract

ZMAT2 is among the least-studied of mammalian proteins and genes, even though it is the ortholog of Snu23, a protein involved in pre-mRNA splicing in yeast. Here we have used data from genomic and gene expression repositories to examine the *Zmat2* gene and locus in 8 terrestrial vertebrates, 10 ray-finned fish, and 1 lobe-finned fish representing > 500 million years of evolutionary diversification. The analyses revealed that vertebrate *Zmat2* genes are similar to their mammalian counterparts, as in 16/19 species studied they contain 6 exons, and in 18/19 encode a single conserved protein. However, unlike in mammals, no Zmat2 pseudogenes were identified in these vertebrates, although an expressed *Zmat2* paralog was characterized in flycatcher that resembled a DNA copy of a processed and retro-transposed mRNA, and thus could be a proto-pseudogene captured during its evolutionary journey from active to inert. The *Zmat2* locus in terrestrial vertebrates, and in spotted gar and coelacanth, also shares additional genes with its mammalian counterparts, including *Histidyl-tRNA synthetase* (*Hars*), *Hars2*, and others, but these are absent from the *Zmat2* locus in teleost fish, in which *Stem-loop-binding protein 2* (*Slbp2*) and *Lymphocyte cytosolic protein 2a* (*Lcp2a*) are present instead. Taken together, these observations argue that a recognizable *Zmat2* was present in the earliest vertebrate ancestors, and postulate that during chromosomal tetraploidization and subsequent re-diploidization during modern teleost evolution, the duplicated *Zmat2* gene was retained and the original lost. This study also highlights how information from genomic resources can be leveraged to reveal new biologically significant insights.

## Introduction

Recent advances in genomics and genetics now provide many opportunities for developing new insights into comparative physiology, evolution, and even disease predisposition [1–3] through the evaluation and interpretation of data found in public genomic and gene expression resources [4]. Yet, among the more than 20,000 protein coding genes in mammalian and in other vertebrate genomes, fewer than 10% have been studied in any detail [5–7]. This is true

**Data Availability Statement:** All relevant data are within the paper and its Supporting Information files.

**Funding:** National Institute of Health research grant, R01 DK042748-28.

**Competing interests:** The author declares that he has no competing interests.

**Abbreviations:** *Dnd1*/*DND1*, dead end microRNA-mediated repression inhibitor 1; *Fam53c*, family with sequence similarity 53, member C; *Fox13a*, forkhead box 13a; *Foxl1-ema-like*, forkhead box L1 epithelial membrane antigen-like; *Hars*/*HARS*, *histidyl-tRNA synthetase*; *Hars2*/*HARS2*, *histidyl-tRNA synthetase* 2; *Ik*/*IK*, IK cytokine; *Lcp2*, lymphocyte cytosolic protein 2; *Lcp2a*, lymphocyte cytosolic protein 2a; NCBI, National Center for Biotechnology Information; *Pcdh1*, protocadherin 1; *Pcdh10*, protocadherin 10; *Pcdh2aa1*, protocadherin 1-alpha-av1; *Pcdha*/*PCDHA*, protocadherin alpha; *Pcdha4*, protocadherin alpha 4; *Pcdhac1*, protocadherin alpha subfamily C1; *Pcdhg*, protocadherin gamma; *Slbp2*, stem-loop binding protein 2; SRA, Sequence Read Archive; *Wdr55*/*WDR55*, WD repeat domain 55; ZMAT, Zinc fingers matrin-type.

even though the ready availability of a variety of genomic, gene-expression, and other databases [4] means that nearly any gene can be analyzed from its smallest molecular features to the physiological, disease-linked, and even population levels [1–3,8–10]. Part of the reasons why most published research has focused on just a small fraction of available genes may be related to historical or social aspects of the biological sciences, or may reflect choices driven by the potential for higher profile publications by association with human diseases or by the availability of specific experimental models that resemble human physiological or pathological processes, and thus may be more amenable for obtaining grant funding [6,7]. Nevertheless, with the vast majority of genes being under-studied or unstudied, the potential for new discoveries of significance is high.

To test this idea, we recently evaluated the *Zmat2* gene in 18 mammalian species, [11]. Mammalian ZMAT2 had been essentially unstudied until a single publication in 2018 focused on its actions in human keratinocyte differentiation [12]. The lack of interest in the protein and in its gene is surprising, since Snu23, the corresponding molecule in yeast, plays a central role in the spliceosome [13], the critical molecular machine in eukaryotes that removes introns from primary gene transcripts [14]. Although human ZMAT2 also has been found to be a component of the spliceosome [15], even this fact has generated little interest in either the protein or gene.

Our investigations found that in addition to the strong conservation of the protein and of protein-coding exons, a remarkable feature about mammalian *Zmat2* is the presence of pseudogenes in half of the species that we have examined to date [11]. Moreover, based on phylogenetic analysis, it appears that these pseudogenes are relatively recent additions to their respective genomes, and have each arisen independently of one another during speciation [11].

The focus of this manuscript is on *Zmat2* in non-mammalian vertebrates. Results show conservation of gene structure, including similarity of protein-coding exons and ZMAT2 proteins during more than 500 Myr of vertebrate evolutionary diversification, but divergence of locus components between teleost fish and other species. Collectively, the data support the idea that this gene is phylogenetically ancient. The observations further demonstrate how investigation and analysis of information found in public genetic and genomic resources can reveal new insights of potentially high biological significance.

## Materials and methods

### Database searches and analyses

The Ensembl Genome Browser (https://useast.ensembl.org/) was examined with BlastN under normal sensitivity (maximum e-value of 10; mis-match scores: 1,-3; gap penalties: opening 5, extension, 2; filtered low complexity regions, and repeat sequences masked) using as queries chicken *Zmat2* DNA fragments (*Gallus gallus*, genome assembly GRCg6a) for terrestrial vertebrates, and zebrafish *Zmat2* for fish (*Danio rerio*, genome assembly GRCz11). Genome assemblies were evaluated for the following species: Amazon molly (*Poecilia formosa*, Poecilia_Formosa-5.1.2), Anole lizard (*Anolis carolinensis*, AnoCar2.0), Chinese softshell turtle (*Pelodiscus sinensis*, PelSin_1.0), cod (*Gadus morhua*, gadMor1), coelacanth (*Latimeria chalumnae*, LatCha1), duck (*Anas platyrhynchos*, CAU_duck1.0), flycatcher (*Ficedula albicollis*, FicAlb_1.4), frog (*Xenopus tropicalis*, JGI 4.2), fugu (*Takifugu rubripes*, FUGU5), Japanese medaka HdrR (*Oryzias latipes*, ASM223467v1), platyfish (*Xiphophorus maculatus*, X_maculatus-5.0-male), spotted gar (*Lepisosteus oculatus*, LepOcu1), stickleback (*Gasterosteus aculeatus*, BROAD S1), tetraodon (*Tetraodon nigroviridis*, TETRAODON 8.0), tilapia (*Oreochromis niloticus*, Orenil1.0), turkey (*Meleagris gallopavo*, Turkey_2.0.1), and zebra finch (*Taeniopygia*

*guttata*, taeGut3.2.4). The highest scoring results in all cases mapped to the *Zmat2* gene, or in flycatcher, to *Zmat2* and to a closely related gene. Additional searches were conducted using as queries portions of *Zmat2* cDNA sequences to follow-up, verify, or extend initial results. The following *Zmat2* cDNAs were obtained from the National Center for Biotechnology Information (NCBI) nucleotide database: Amazon molly (accession number: XM_007577084), chicken (NM_001031465), Chinese softshell turtle (XM_014579525), fugu (XM_003970852), medaka (XM_004076042), platyfish (XM_005811572), spotted gar (XM_006631892), tilapia (XM_003457022), turkey (XM_019620338), zebra finch (XM_030283776), and zebrafish (NM_212681, BC097020). ZMAT2 protein sequences were obtained from the Uniprot browser (http://www.uniprot.org/); in the absence of primary protein data, DNA sequences of *ZMAT2* exons were translated using Serial Cloner 2.6 (see: http://serialbasics.free.fr/Serial_Cloner.html).

## Mapping the 5' and 3' ends of *Zmat2* genes

Inspection of *Zmat2* and its proposed mRNAs in the Ensembl genome database for most species revealed lack of either 5' or 3' untranslated regions (UTR) for its mRNAs. For all species except for platyfish, RNA-sequencing libraries found in the NCBI Sequence Read Archive (SRA; www.ncbi.nlm.nih.gov/sra) were queried with multiple 60 base pair probes from genomic DNA corresponding to presumptive 5' segments of exon 1, and from 3' portions of exon 6, and read counts were analyzed. All queries used the Megablast option (optimized for highly similar sequences; maximum target sequences–1,000 (this parameter may be set from 50 to 20,000); expect threshold–10; word size–11; match/mismatch scores–2, -3; gap costs–existence 5, extension 2; low-complexity regions filtered).

## DNA and protein alignments

Multiple sequence alignments were performed for ZMAT2 proteins from different species. DNA sequences were uploaded into the command line of Clustalw2 (https://www.ebi.ac.uk/Tools/msa/clustalw2/) [16] in FASTA format. Clustalw2 first performs pairwise sequence alignments using a progressive alignment approach, after which it creates a guide tree using a neighbor joining algorithm, which is then used to complete a multiple sequence alignment. Pairwise alignments comparing ZMAT2 proteins discovered in *Xenopus tropicalis*, and comparing ZMAT2 proteins from the two *Zmat2* genes in flycatcher were performed using Needle (EMBOSS), which creates an optimal global alignment of two sequences using the Needleman-Wunsch algorithm [16].

## Results

### Characterizing the chicken *Zmat2* gene

Human *ZMAT2* and other mammalian *Zmat2* genes had been incompletely characterized in the Ensembl and UCSC genomic repositories, but as shown in our recent studies, they comprise a highly conserved series of orthologous genes and proteins [11]. In 17 of 18 mammalian genomes analyzed, *Zmat2* was found to be a single copy gene, with the single exception being a tandem duplication identified in the opossum genome. As *Zmat2* was present in a single copy in other monotremes and marsupials, the duplication event most likely occurred after the divergence of these species from each other [11]. We also determined that the genomes of nine of the mammals studied contained one or more *Zmat2* pseudogenes, and that based on phylogenetic analysis, the insertion of each of these pseudogenes into each genome occurred after

the divergence of each individual mammal from its nearest ancestor [11]. These observations were the impetus to examine *Zmat2* in non-mammalian vertebrates.

Chicken *Zmat2* was chosen as the reference gene for terrestrial vertebrates, primarily because it has been more extensively studied than other birds, reptiles, or amphibians. According to data in Ensembl as of January 2020, single-copy chicken *Zmat2* consists of 6 exons on chromosome 13, with both exons 1 and 6 containing untranslated regions (UTRs), and exons 2 through 5 being composed entirely of coding information. As in mammals, chicken *Zmat2* is located adjacent to the *Hars2* and *Hars* genes (Fig 1A) [11]. The NCBI nucleotide database contained a single experimentally defined chicken *Zmat2* cDNA, but this sequence had shorter 5' and 3' UTRs than described for the genomic sequence, and thus was not useful for additional gene analysis.

Direct mapping of the coordinates of the chicken *Zmat2* gene that are expressed in an RNA-sequencing library from liver found in the NCBI SRA (see S1 Table) revealed that exon 1 contained a 5' UTR of at least 89 base pairs, which was 54 base pairs longer than described in Ensembl (Fig 1B). However, no potential TATA boxes or initiator elements, which position RNA polymerase II at the start of transcription [17,18], were found adjacent to the 5' end of this transcript. Thus, the beginning of the chicken *Zmat2* gene is at best tentatively defined.

Analogous studies using probes from different parts of chicken *Zmat2* exon 6 demonstrated that this exon was 1903 base pairs in length, which is 35 nucleotides longer than stated in Ensembl (Fig 1B). The 3' end of the exon contained a presumptive poly A recognition sequence of 'ATTTTA', and a poly A addition site [19] was mapped 8 base pairs further 3' (Fig 1B). Collectively, these results describe a 6-exon chicken *Zmat2* gene of approximately 9159 base pairs in length (Fig 1C) that is transcribed and processed into a 2452 nucleotide mRNA (Fig 1D), and that encodes a 199-amino acid ZMAT2 (Fig 1E). Therefore, chicken *Zmat2* and its locus resembles the *Zmat2* genes of other mammals [11].

## The *Zmat2* gene in other terrestrial vertebrates

By using as queries chicken *Zmat2* exons, along with annotated data from Ensembl, and in selected cases, homologous cDNAs or exons from more closely related species, *Zmat2* also appeared to be a 6-exon gene in turkey, duck and zebra finch, and a 5-exon gene in flycatcher and Chinese softshell turtle, although in most of these species the annotated genomic data in Ensembl were incomplete (e.g., no 5' UTR in exon 1 in turkey or zebra finch, and no 3' UTR in the last Chinese softshell turtle exon; Table 1). Mapping experiments using gene expression libraries from the NCBI SRA data resource were able to fill in much of the missing information (see S1 Fig, Fig 2, Table 2). However, an exon 1 similar to that of other species could not be identified in flycatcher or in Chinese softshell turtle (Table 1).

The outliers among terrestrial vertebrate *Zmat2* genes appeared to be Anole lizard and the tropical clawed frog, *Xenopus tropicalis*. The former was described in Ensembl to contain 7 exons, the 6 described for chicken *Zmat2* plus an additional 5' exon of 944 base pairs (Table 1). However, expression of *Zmat2* transcripts containing this 'extra' exon were not detected in liver or in other tissues after screening RNA-sequencing libraries with appropriate DNA probes. Thus, the tentative conclusion is that this exon is not part of the Anole lizard *Zmat2* gene.

Frog *Zmat2* was described as having 8 exons in Ensembl (Table 2, Fig 3). Mapping studies with a gene expression library from liver (see S1 Table) revealed that exon 1 was 385 nucleotide pairs in length, and that exon 8 consisted of 1956 base pairs (Table 2, Fig 3B). Moreover, the 3' end of exon 8 contained a presumptive poly A recognition sequence of 'AATAAA', and tandem poly A addition sites (Fig 3B). Transcripts containing the additional exons (#5, 6, 7) also were

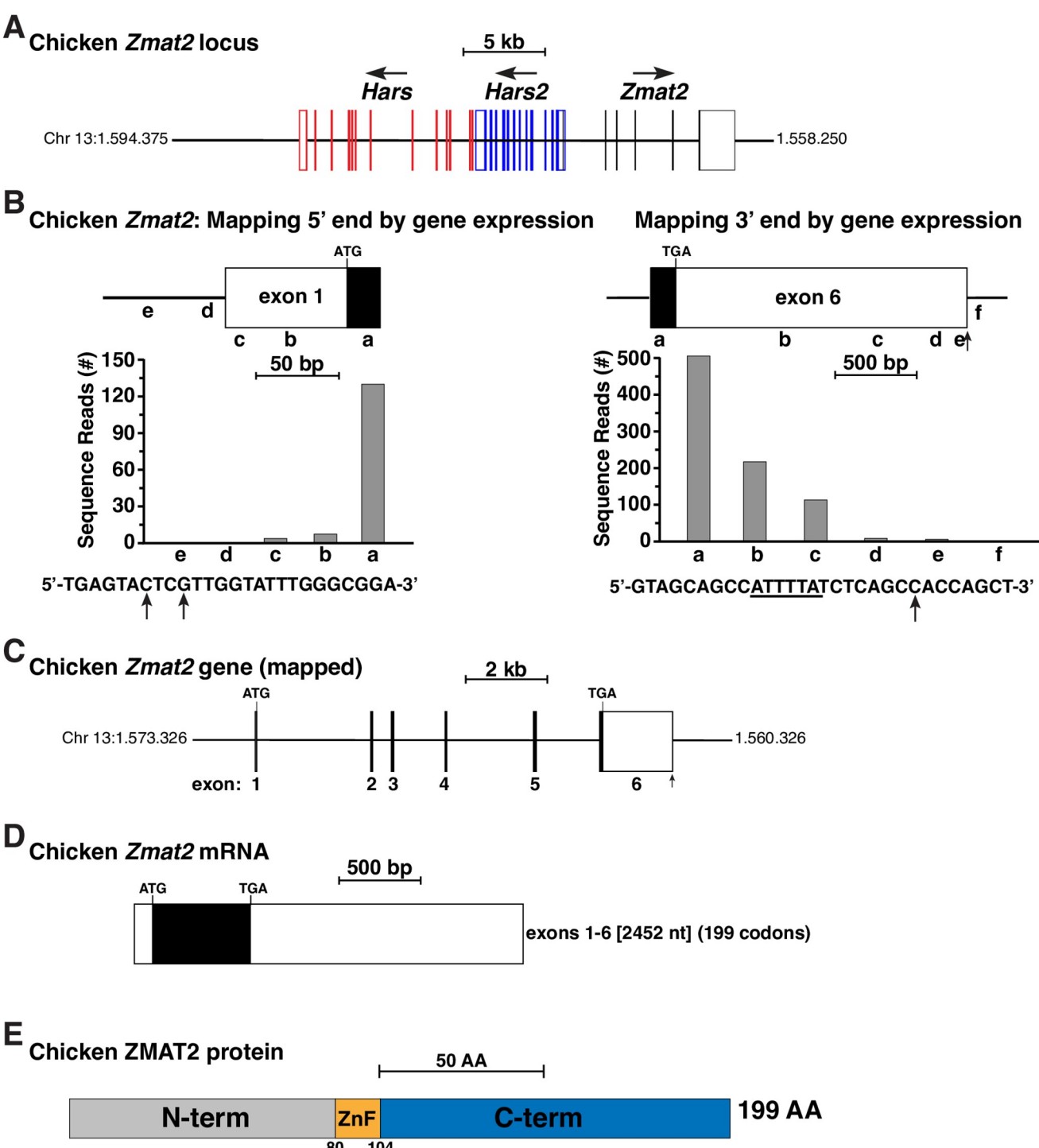

**Fig 1. Organization of the chicken *Zmat2* locus and gene. A**. Diagram of the chicken *Zmat2* locus and gene on chromosome 13. Exons are depicted by lines and boxes (red for *Hars*, blue for *Hars2*, black for *Zmat2*), with coding regions solid and non-coding segments white. The direction of transcription is indicated for each gene and a scale bar is shown. **B**. Mapping the beginning and end of chicken *Zmat2*: diagram of exon 1 (left) and exon 6 (right), and graphs of gene expression data from the NCBI SRA RNA-sequencing library, SRX3729588 (S1 Table), using 60 base pair genomic segments a-e, and a-f, respectively, as probes. The DNA sequence below the left graph depicts the putative start of exon 1, with the locations of the 5' ends of the longest RNA-sequencing clones indicated by vertical arrows. Below the right graph is the DNA sequence of the putative 3' end of exon 6. A potential polyadenylation signal (ATTTTA) is underlined and a vertical arrow denotes the possible poly A addition site. **C**. Map of the chicken *Zmat2* gene, including ATG and TGA codons, and the 5' and 3' ends as defined in **B**. Coding regions are in black and noncoding portions in white. A scale bar is shown. **D**. Diagram of the chicken *Zmat2* mRNA, with coding regions in black and non-coding segments in white. The length is indicated in nucleotides (nt) as are the number of codons. **E**. Schematic of the chicken ZMAT2 protein, with NH_2 (N), COOH (C) terminal (term), and zinc finger (ZnF) regions labeled and color-coded.

**Table 1. Terrestrial vertebrate *Zmat2* genes in Ensembl.**

| Species | Exon 1 5' UTR (nt) | Exon 1 coding (nt) | Exon 6 coding (nt) | Exon 6 3'UTR (nt) |
|---|---|---|---|---|
| Chicken | 35 | 18 | 144 | 1724 |
| Turkey | None | 18 | 144 | 678 |
| Duck | 875 | 21 | 144 | 688 |
| Zebra finch | None | 18 | 144 | 438 |
| Flycatcher* | None | None | 144 | 1681 |
| Ch softshell turtle | None | None | None | None |
| Anole lizard | #154 | #18 | ^144 | ^2675 |
| Xen. tropicalis | None | 18 | 144 | None |

*gene name is ENSFALT00000009421.1;

#exon 2;

^exon 7

found in *Zmat2* mRNAs in several different frog tissues (Fig 3C), even though orthologs of these exons were not detected in any other terrestrial vertebrate species. Moreover, of note, there is no cognate of exon 5 from other vertebrates in *Xenopus tropicalis Zmat2*. Collectively, the different *Zmat2* exons were transcribed and processed into 3 classes of mRNAs that encoded *Xenopus* ZMAT2 proteins of 172, 197, and 198 amino acids (Fig 3D and 3E). Thus, *Xenopus tropicalis Zmat2* appears to be the outlier among terrestrial vertebrates.

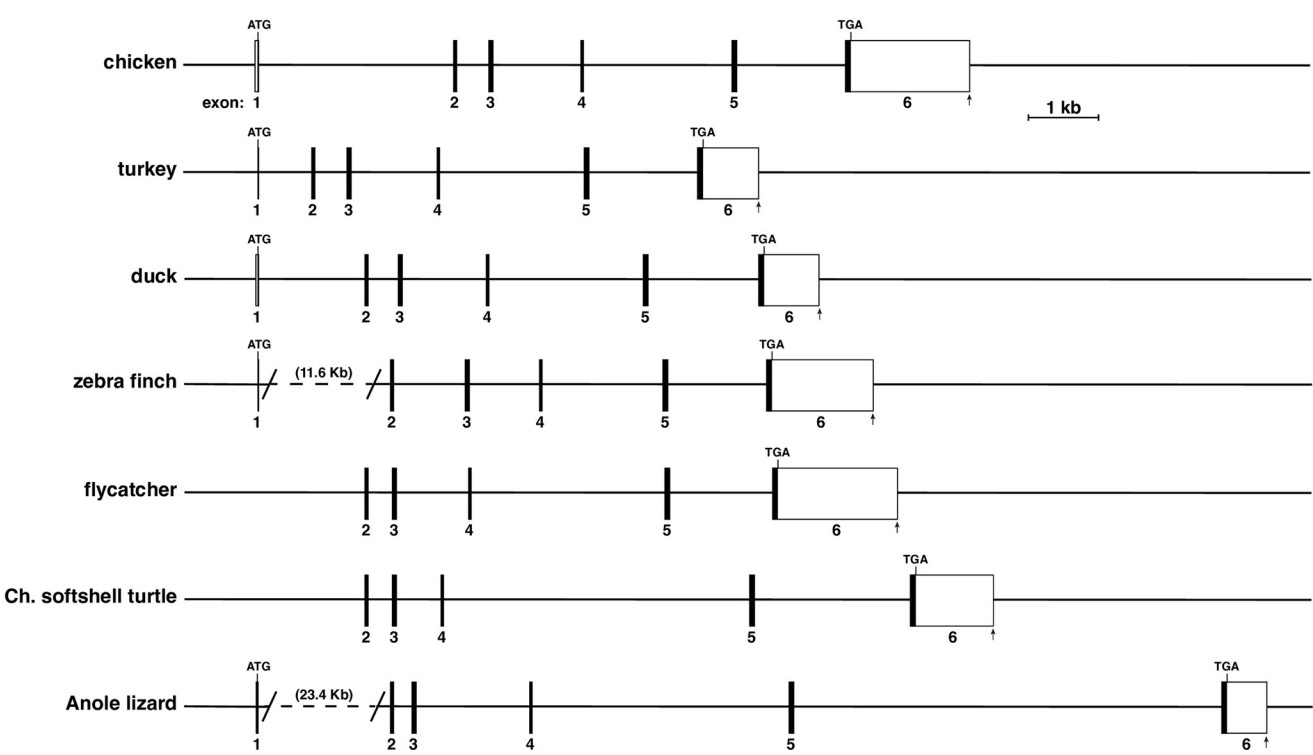

**Fig 2. The *Zmat2* gene in terrestrial vertebrates.** Diagrams of chicken, turkey, duck, zebra finch, flycatcher, Chinese (Ch.) softshell turtle, and Anole lizard *Zmat2* genes. Exons are indicated as boxes, with coding regions in black and non-coding segments in white for each gene. The locations of ATG and TGA codons are indicated, as is the putative 3' end of exon 6 for each gene (vertical arrow). A scale bar is shown. See Tables 2 and 3 for more details.

**Table 2. Characterization of terrestrial vertebrate *Zmat2* genes (in base pairs).**

| Species | Exon 1 | Intron 1 | Exon 2 | Intron 2 | Exon 3 | Intron 3 | Exon 4 | Intron 4 | Exon 5 | Intron 5 | Exon 6 | Total Length |
|---|---|---|---|---|---|---|---|---|---|---|---|---|
| Chicken | 107 | 2807 | 94 | 558 | 124 | 1345 | 74 | 2338 | 146 | 1532 | 1903 | 9159 |
| Turkey | 18 | 807 | 94 | 566 | 124 | 1317 | 74 | 2277 | 146 | 1507 | 823 | >7753 |
| Duck | 82 | 1514 | 94 | 541 | 124 | 1237 | 74 | 2503 | 146 | 1582 | 814 | 8723 |
| Zebra finch | 18 | 11613 | 94 | 1092 | 124 | 1036 | 74 | 1724 | 146 | 1466 | 1555 | >18202 |
| Flycatcher | nd | nd | 94 | 450 | 124 | 1016 | 74 | 2740 | 146 | 1468 | 1766 | >6873 |
| Ch. softshell turtle | nd | nd | 94 | 467 | 124 | 654 | 74 | 4303 | 146 | 2225 | 1150 | >9237 |
| Anole lizard | 48 | 23430 | 94 | 241 | 124 | 1608 | 74 | 3656 | 146 | 6011 | 646 | 36078 |

| Species | Ex 1 | In 1 | Ex 2 | In 2 | Ex 3 | In 3 | Ex 4 | In 4 | Ex 5 | In 5 | Ex 6 | In 6 | Ex 7 | In 7 | Ex 8 | Total Length |
|---|---|---|---|---|---|---|---|---|---|---|---|---|---|---|---|---|
| X tropicalis | 385 | 4263 | 94 | 2274 | 124 | 4263 | *122 | 982 | 46 | 82 | 46 | 1505 | 91 | 1274 | 1956 | 17507 |

*126, 139 base pairs in other mRNAs

nd—not detected

DNA similarity with chicken *Zmat2* exon 1 could not be established among the other terrestrial vertebrates studied here (Table 3). In contrast, coding exons 2–5 were identical in length (Table 2) and highly conserved among all species (82.9% to 100% DNA identity, Table 3). Exon 6 also was fairly well conserved, including part of the 3' UTR (84.6% to 95.6% identity, Table 3). When all of this information was aggregated, gene length varied among fully-characterized terrestrial vertebrate *Zmat2* genes over a 4-fold range, from 8723 base pairs in duck to 36078 base pairs in Anole lizard, with most of the differences attributable to introns, especially intron 1 (Table 2, Fig 2). Of note, the flycatcher and Chinese softshell turtle *Zmat2* genes could not be accurately sized because of lack of identification of an exon 1 and intron 1 (Table 2).

## The flycatcher genome contains an expressed *Zmat2* 'pseudogene'

Screening the flycatcher genome with chicken *Zmat2* exons led to the identification of two sets of related DNA sequences, with the second group having levels of similarity with chicken *Zmat2* exons 2 through 6 ranging from 84.2% to 94.6% (Fig 4A–4C). These latter DNA sequences were adjacent to one another (Fig 4A–4C), and thus resembled a pseudogene that had been derived from an mRNA that had been retro-transposed back into the flycatcher genome as a DNA copy [20]. However, unlike pseudogenes, flycatcher *Zmat2-2* was expressed (Fig 4B and 4D), although at mRNA levels that were 2.5%–38% of *Zmat2-1* in different organs and tissues (Fig 4D). Perhaps surprisingly, *Zmat2-2* encoded an orthologue of exon 1, with a predicted 6-residue amino acid sequence (MASGSG) that matched the $NH_2$-terminus of the chicken protein (Fig 4E). The two ZMAT2 proteins in flycatcher were only 89.6% identical to one another (173/193 amino acids; Fig 4E), with ZMAT2-1 being more similar than ZMAT2-2 to the chicken protein (Table 4).

## ZMAT2 protein sequences are less conserved in terrestrial vertebrates than among mammals

In mammals, 199-residue ZMAT2 was identical to the mouse protein in 10 of 17 other species studied, and in all eight species with altered amino acids, only 1 or 2 changes were found (>99% identity) [11]. Among the terrestrial vertebrates examined here, none of the proteins were identical to each other (Table 4, Fig 5), or to human or mouse ZMAT2, and multiple amino acid differences were found in inter-species comparisons (Table 4, Fig 5). Moreover, protein length in terrestrial vertebrates ranged from 172 amino acids (frog ZMAT2-3) to 201

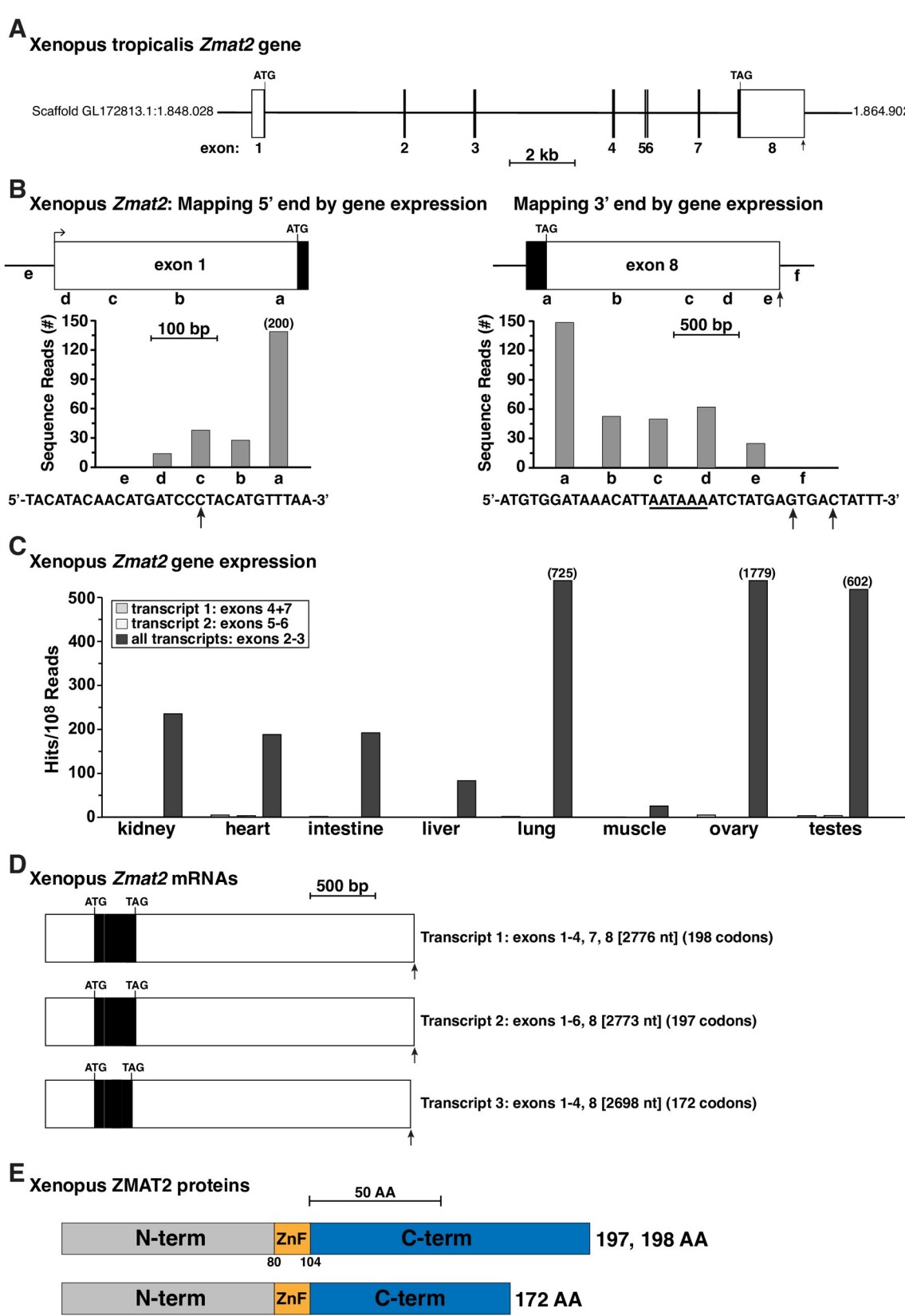

**Fig 3. The *Xenopus tropicalis Zmat2* gene is more complicated than in other terrestrial vertebrates. A**. Diagram of the *Xenopus tropicalis Zmat2* gene. Exons are depicted as boxes, with coding segments solid and noncoding regions white. The 5' end of exon 1 and the 3' end of exon 8 were defined in **B**. The locations of ATG and TAG codons are indicated, and a scale bar is shown. **B**. Mapping the

beginning and end of *Xenopus tropicalis Zmat2*: diagram of exon 1 (left) and exon 8 (right), and graphs of gene expression data from the NCBI SRA RNA-sequencing library, SRX6631062 (Additional S1 Table), using 60 base pair genomic segments a-e, and a-f, respectively, as probes. The arrow in the DNA sequence below the left graph depicts the putative 5' end of exon 1. Below the right graph, the putative 3' end of exon 6 is illustrated, with a potential polyadenylation signal (AATAAA) underlined and two possible poly A addition sites marked by vertical arrows. **C**. Gene expression data from tissue-specific NCBI SRA RNA-sequencing libraries (see Additional S1 Table) for different *Zmat2* mRNA species, using probes for exons 4 + 7, 5–6, or 2–3 (Additional S2 Table) that discriminate among transcripts. **D**. Diagram of *Xenopus tropicalis Zmat2* transcripts. The mRNA species vary by the inclusion of different exons. The coding segments are in black and non-coding regions in white. Lengths are in nucleotides and the number of codons in each open reading frame is listed. **E**. Diagram of ZMAT2 proteins, with NH$_2$ (N), COOH (C) terminal (term), and zinc finger (ZnF) regions labeled and color-coded.

residues (duck), although it consisted of 199 amino acids in 5 species, chicken, turkey (but no NH$_2$-terminal methionine residue), zebra finch, flycatcher (ZMAT2-2 only; in which there was an amino acid change in the zinc finger domain), and Anole lizard (Table 4, Fig 5).

## Characterizing *Zmat2* genes in fish

Zebrafish was chosen as the index species for examining *Zmat2* genes in fish, as it has been more extensively studied than other teleost or non-teleost species. Based on information in Ensembl as of January 2020, the zebrafish genome encodes a single *Zmat2* gene on chromosome 21 located adjacent to and in the opposite transcriptional orientation to *Slbp2* (and not *Hars2* as in chicken or in mammals [11] (compare Fig 6A with Fig 1A). According to Ensembl, zebrafish *Zmat2* has six exons, and both exons 1 and 6 have untranslated regions (UTRs), while exons 2 through 5 are composed of coding information (Table 5). The NCBI nucleotide database contained a single zebrafish *Zmat2* cDNA, but this was not useful for additional gene characterization at either the 5' or 3' ends.

Mapping of the 5' end of the zebrafish *Zmat2* gene using RNA-sequencing libraries from heart and liver (S1 Table), demonstrated that exon 1 contained a 5' UTR of 18 base pairs, (Fig 6C), 80 base pairs shorter than stated in Ensembl (Table 5). As in chicken *Zmat2*, no potential TATA boxes or initiator elements [17,18] could be found 5' to the longest transcript identified in these libraries (Fig 6C). Thus, the start of zebrafish *Zmat2* gene has not been definitively established.

Mapping the 3' end of exon 6 with the same RNA-sequencing libraries led to identification of a 3' UTR of 499 base pairs, and a total exon length of 643 base pairs, which included near its 3' end a 'ATTAAA' presumptive poly A recognition sequence and 12 base pairs further 3', a

**Table 3. Nucleotide identity with chicken *Zmat2* exons.**

| Species | Exon 1 (107 bp)* | Exon 2 (94 bp) | Exon 3 (124 bp) | Exon 4 (74 bp) | Exon 5 (146 bp) | Exon 6 (1903 bp)* |
|---|---|---|---|---|---|---|
| Turkey | no match | 98.9 | 98.4 | 100 | 100 | 95.6 (823 bp) |
| Duck | no match | 96.8 | 96.0 | 97.3 | 97.3 | 89.5 (814 bp) |
| Zebra finch | no match | 93.4 | 96.8 | 96.0 | 95.2 | 85.0 (1555 bp) |
| Flycatcher | no match | 93.4 | 95.2 | 91.9 | 93.8 | 84.6 (1617 bp) |
| Ch. softshell turtle | no match | 91.5 | 92.0 | 91.6 | 95.1 | 87.8 (378 bp) |
| Anole lizard | no match | 90.3 | 89.1 | 90.4 | 88.6 | 89.8 (284 bp) |
| Xen tropicalis | no match | 86.6 | 82.9 | 87.5 | no exon | #no match |

*coding and non-coding DNA;

#exon 8

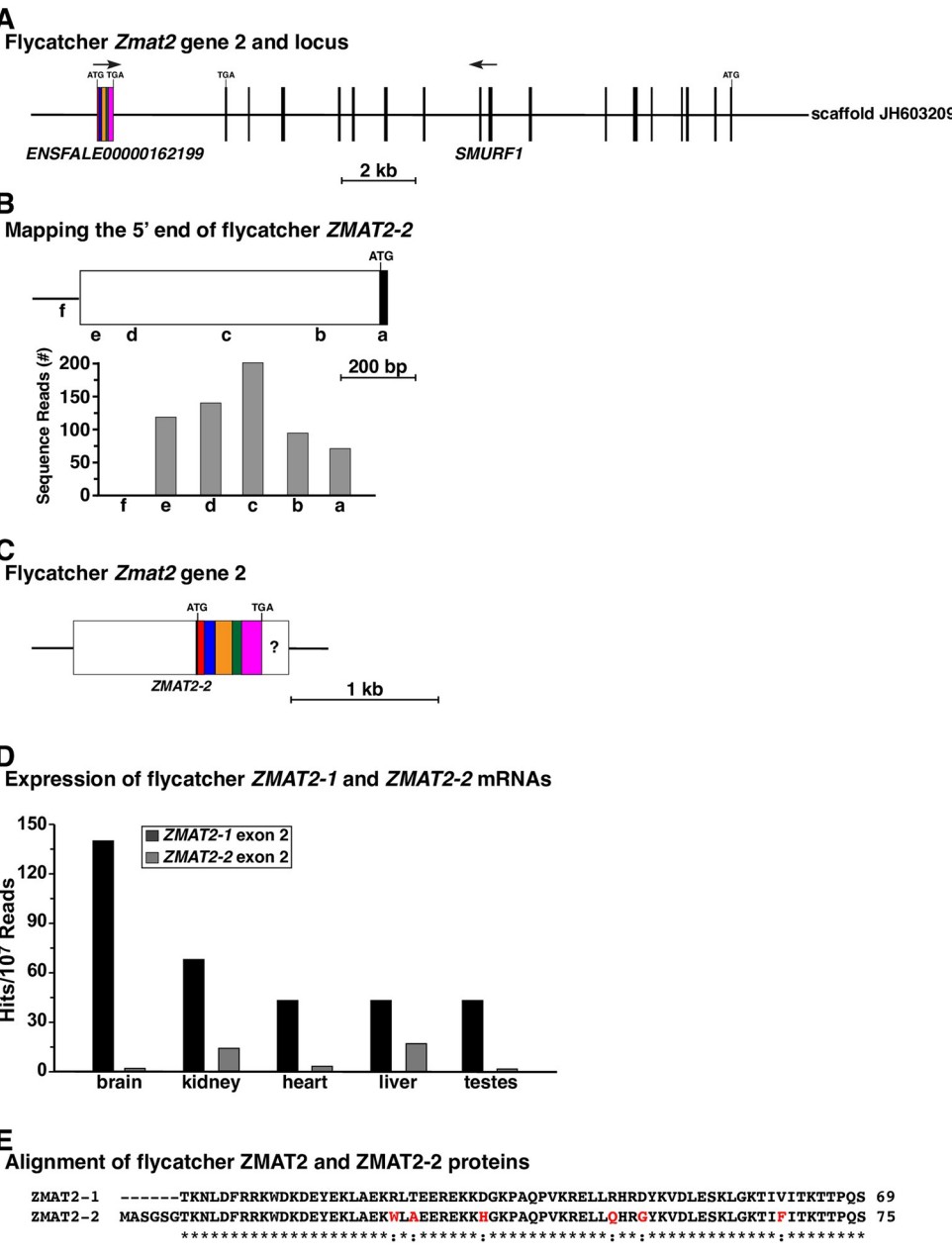

**Fig 4. The flycatcher genome contains a second *Zmat2* gene. A**. Schematic of flycatcher *Zmat2* gene 2 and its location adjacent to *Smurf1*. Locations of ATG and TGA codons are indicated, and a scale bar is shown. Color-coding indicates regions that are similar in DNA sequence to individual exons of *Zmat2* gene 1. **B**. Mapping the 5' end of exon 1 of *Zmat2* gene 2. The diagram illustrates exon 1, with the graph below showing gene expression data from NCBI SRA RNA-sequencing library, SRX6380621 (Additional S1 Table), using 60 base pair genomic segments a-f as probes. **C**. Diagram of flycatcher *Zmat2-2* mRNA, with coding regions in different colors and non-coding segments in white. The question mark indicates that the 3' end could not be mapped because of identity with *Zmat2-1*. **D**. Gene expression data for both flycatcher *Zmat2* mRNA species, using probes from exon 2 of each gene (Additional S2 Table) that discriminate between transcripts. **E**. Alignment of amino acid sequences of flycatcher ZMAT2 proteins. Similarities and differences are shown, with identities being indicated by asterisks, and differences by colons. Differences in ZMAT2-2 also are marked in red.

**Table 4. Amino acid identities with chicken ZMAT2 in terrestrial vertebrates.**

| Species | Length | Percent Identity | Amino Acid Differences |
|---|---|---|---|
| Turkey | 199* | 98.5 | $M^1 > S; S^3 > L; G^4 > S$ |
| Duck | 201 | 97.0 | $M^1 > I; A^2 > C; S^3 > D; G^4 > N; S^5 > A; G^6 > L$ |
| Zebra finch | 199 | 98.0 | $M^1 > F; A^2 > S; G^4 > N; S^5 > T$ |
| Flycatcher | 193# | 99.5 | $N^{198} > S$ |
| Flycatcher 2 | 199 | 90.5 | see Fig 5 |
| Ch. softshell turtle | 193# | 99.0 | $A^{42} > V; N^{198} > S$ |
| Anole lizard | 199 | 98.0 | $T^7 > A; A^{42} > V; T^{176} > A N^{198} > T$ |
| Xen tropicalis 1 | 197 | 75.1 | see Fig 5 |
| Xen tropicalis 2 | 198 | 75.8 | see Fig 5 |
| Xen tropicalis 3 | 172 | 82.0 | see Fig 5 |

*No $NH_2$-terminal methionine;

#$NH_2$-terminus undefined

putative poly A addition site [19,21] (Fig 6C). Taken together, these results define a 6-exon zebrafish *Zmat2* gene of approximately 6404 base pairs in length (Fig 6B) that is transcribed and processed into an mRNA of 1114 nucleotides (Fig 6D), and that encodes a 198-amino acid ZMAT2 protein (Fig 6E). Thus, although more compact, zebrafish *Zmat2* appears organizationally to resemble the chicken (Fig 1C), mouse, and other mammalian genes [11].

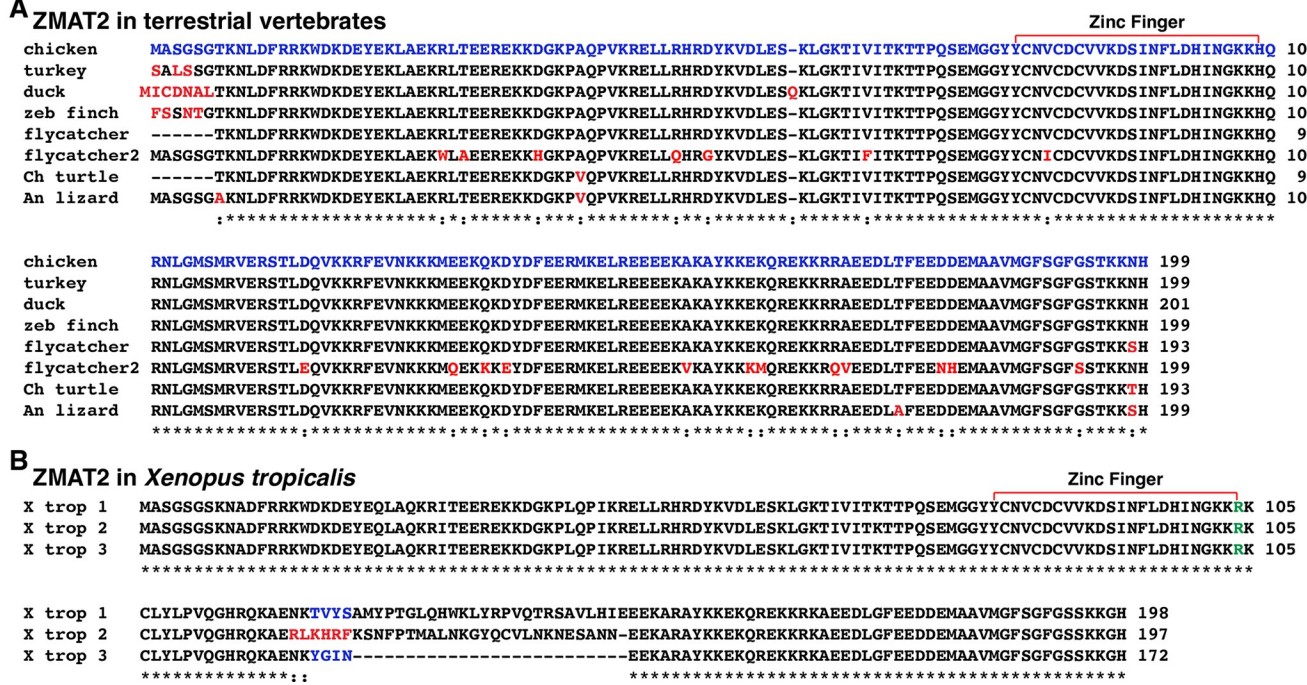

**Fig 5. Terrestrial vertebrate ZMAT2 proteins. A**. Alignments of amino acid sequences of ZMAT2 proteins from different terrestrial vertebrate species are shown in single letter code. Identities and differences among proteins are indicated, with identities labeled by asterisks, and differences by colons. Dashes indicating no residue have been placed to maximize alignments. Red text also depicts differences. The zinc finger region has been labeled. **B**. ZMAT2 in *Xenopus tropicalis*. The three different proteins have been aligned. Dashes have been placed to maximize alignments. Identities are labeled by asterisks, and variant amino acids are depicted by colons and are labeled red and blue. The single change with other vertebrates of histidine to arginine in the zinc finger domain has been labeled in green. See Table 4 for additional information.

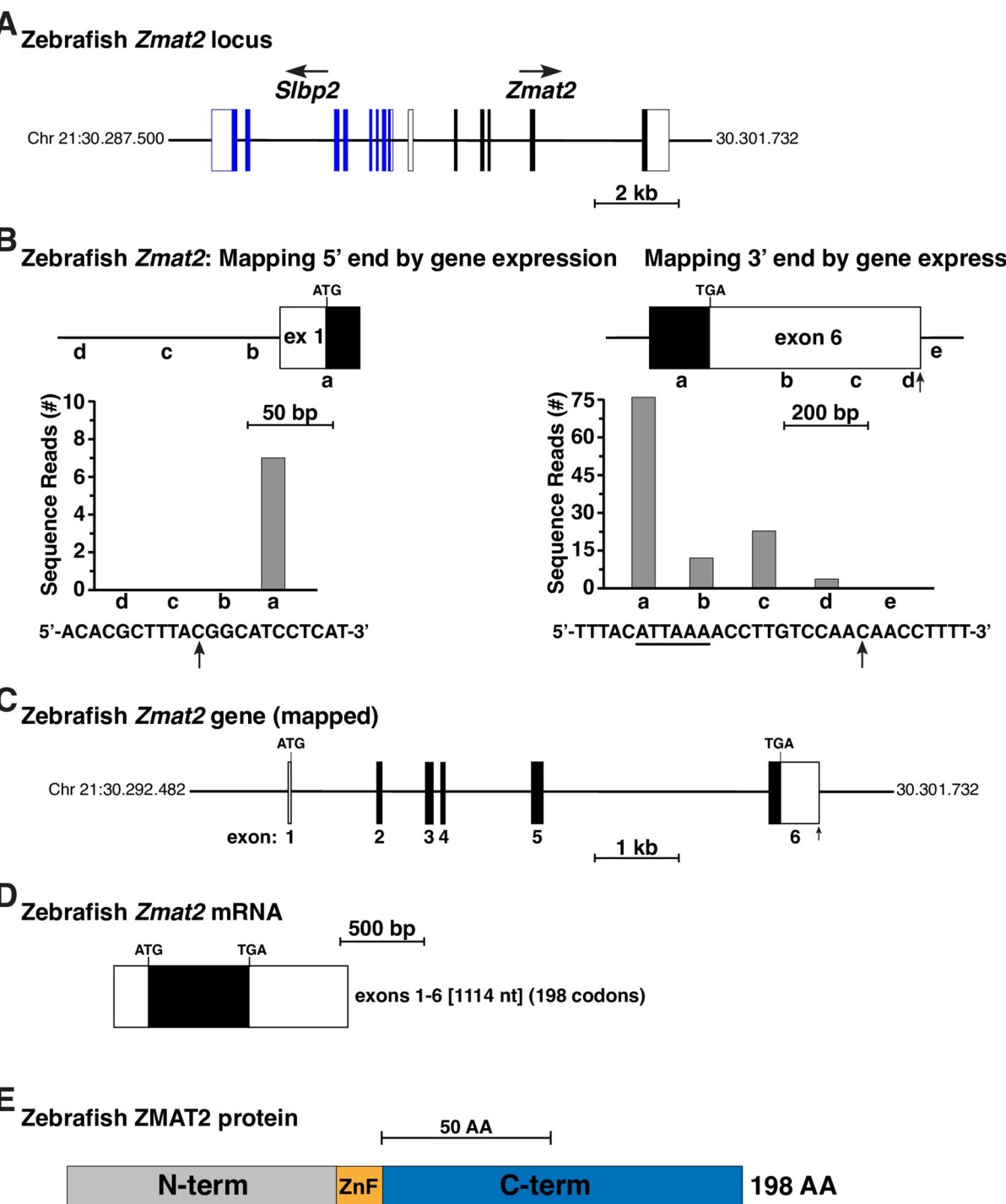

**Fig 6. Organization of the zebrafish *Zmat2* locus and gene. A**. Diagram of the zebrafish *Zmat2* locus and gene on chromosome 21. Exons are depicted by lines and boxes (blue for *Slbp2*, black for *Zmat2*), with coding regions solid and non-coding segments white. The direction of transcription is indicated for each gene and a scale bar is shown. **B**. Mapping the 5' and 3' ends of zebrafish *Zmat2*: diagram of exon 1 (left) and exon 6 (right), and graphs of gene expression data from the NCBI SRA RNA-sequencing library, SRX6603852 (Additional S1 Table), using 60 base pair genomic segments a-d or a-e as probes. The DNA sequence below the left graph depicts the putative start of exon 1, with the 5' end of the longest RNA-sequencing clone indicated by a vertical arrow. The DNA sequence of the putative 3' end of exon 6 is shown below the right graph. A potential polyadenylation signal (ATTAAA) is underlined and a vertical arrow denotes the possible poly A addition site. **C**. Map of the zebrafish *Zmat2* gene, including 5' and 3' ends as delineated in **B**. Coding regions are in black and noncoding portions in white. The locations of ATG and TGA codons are indicated, as is the putative 3' end of exon 6. A scale bar is shown. **D**. Diagram of the zebrafish *Zmat2* mRNA, with coding regions in black and non-coding segments in white. The length is indicated in nucleotides (nt) as are the number of codons. **E**. Schematic of the zebrafish ZMAT2 protein, with NH$_2$ (N), COOH (C) terminal (term), and zinc finger (ZnF) regions labeled and color-coded.

**Table 5. Fish *Zmat2* genes in Ensembl.**

| Species | Exon 1 5' UTR (nt) | Exon 1 coding (nt) | Exon 6 coding (nt) | Exon 6 3'UTR (nt) |
|---|---|---|---|---|
| Tetraodon | None | 18 | 144 | None |
| Fugu | 465 | 18 | 144 | None |
| Stickleback | None | 18 | 144 | None |
| Medaka | 870 | 18 | 144 | 2004 |
| *Cod | None | 18 | None | None |
| #Tilapia | None | 18 | 144 | None |
| Amazon molly | 123 | 18 | 144 | 656 |
| Platyfish | None | 18 | 144 | None |
| Zebrafish | 98 | 18 | 144 | 501 |
| Spotted gar | None | 123 | 144 | 1638 |
| Coelacanth | 105 | 18 | 144 | 1519 |

*5 exons;

#8 exons

## The *Zmat2* gene in other fish

By using zebrafish *Zmat2* exons as queries, along with annotated data from Ensembl, and in selected cases exons from other species, *Zmat2* also appeared to be a 6-exon gene in Amazon molly, fugu, medaka, platyfish, spotted gar, stickleback, tetraodon, coelacanth, and cod (exon 6 was identified with the orthologous region of the Amazon molly *Zmat2* gene), and was stated to be an 8-exon gene in tilapia (Fig 7, Tables 5 and 6). In many of these fish species the genomic data in Ensembl was incomplete (no 5' UTR in exon 1 in tetraodon, stickleback, cod, tilapia, platyfish, and spotted gar, and no 3' UTR in tetraodon, fugu, stickleback, tilapia, and platyfish, Table 5), but the UTRs were able to be mapped in most species by a combination of DNA sequence similarity with zebrafish *Zmat2* and by analysis of gene expression using RNA sequencing libraries (S1 Table, S1 Fig). Other mapping experiments showed that the 'extra' *Zmat2* exons of 24 and 8 base pairs in tilapia identified in Ensembl were not included in any *Zmat2* transcripts detected in liver. Therefore, with the possible exception of tilapia, it appears that *Zmat2* gene structure in fish is similar to other vertebrates and to mammals (Fig 7, Table 6).

DNA conservation with zebrafish *Zmat2* exons 1, 2, 4, and 6 was minimal among most of the 10 teleost and non-teleost species examined here, in contrast to what was discovered for chicken *Zmat2* and terrestrial vertebrates, in which exons 2 through 6 were highly related to one another (compare Tables 3 and 7). In addition, when all of the data were compiled, fish *Zmat2* genes appeared generally to be more compact than in terrestrial vertebrates (compare Tables 2 and 6), although they also varied substantially in gene length, from 2337 base pairs in zebrafish to 16951 base pairs in Amazon molly (Fig 7, Table 6).

## ZMAT2 proteins in fish

Except for zebrafish (198 amino acids), platyfish (196 residues), and coelacanth (199 amino acids), all fish ZMAT2 proteins analyzed here were over 200 residues in length (201 amino acids in tetraodon, fugu, stickleback, medaka, tilapia, and Amazon molly; 202 residues in cod, and 225 amino acids in spotted gar, with the extra residues in the latter being located at the $NH_2$-terminus (Fig 8, Table 8)). In all of these species, the 23-amino acid zinc finger domain was identical, and the majority of differences were located within the $NH_2$-terminal segment of the protein (Fig 8).

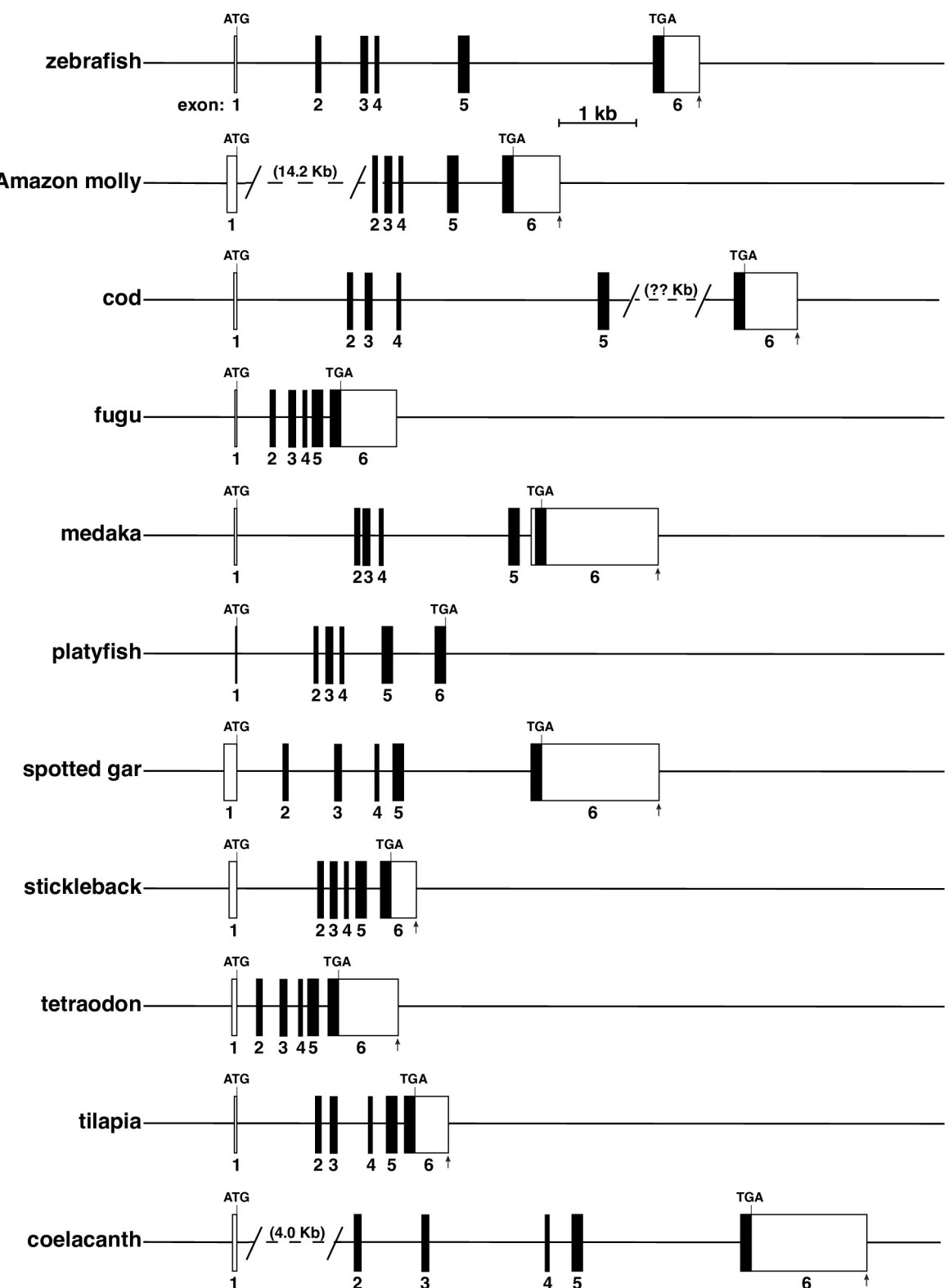

**Fig 7. The *Zmat2* gene in fish.** Diagrams of zebrafish, Amazon molly, cod, fugu, medaka, platyfish, spotted gar, stickleback, tetraodon, tilapia, and coelacanth *Zmat2* genes. Exons are indicated as boxes, with coding regions in black and non-coding segments in white for each gene. Locations of ATG and TGA codons are indicated, as is the putative 3' end of exon 6 for each gene (vertical arrow). A scale bar is shown. See Tables 6 and 7 for more details.

**Table 6. Characterization of fish *Zmat2* genes (in base pairs).**

| Species | Exon 1 | Intron 1 | Exon 2 | Intron 2 | Exon 3 | Intron 3 | Exon 4 | Intron 4 | Exon 5 | Intron 5 | Exon 6 | Total Length |
|---|---|---|---|---|---|---|---|---|---|---|---|---|
| Tetraodon | 71 | 494 | 100 | 213 | 124 | 130 | 74 | 69 | 146 | 100 | 928 | 2337 |
| Fugu | 28 | 477 | 100 | 298 | 124 | 87 | 74 | 67 | 146 | 114 | 945 | 2460 |
| Stickleback | 110 | 1075 | 100 | 88 | 124 | 84 | 74 | 92 | 146 | 180 | 431 | 2504 |
| Medaka | 38 | 1498 | 100 | 85 | 124 | 128 | 74 | 1665 | 146 | 172 | 1776 | 5806 |
| *Cod | 43 | 1338 | 103 | 214 | 124 | 329 | 74 | 2740 | 146 | unknown | 881 | unknown |
| Tilapia | 38 | 989 | 100 | 113 | 124 | 371 | 74 | 162 | 146 | 80 | 611 | 8614 |
| Amazon molly | ^141 | 14216 | 100 | 114 | 124 | 77 | 74 | 603 | 146 | 561 | 795 | 16951 |
| #Platyfish | >18 | 957 | 85 | 114 | 124 | 77 | 74 | 585 | 149 | 558 | >144 | >2885 |
| Zebrafish | 36 | 978 | 91 | 503 | 124 | 85 | 74 | 1040 | 146 | 2684 | 643 | 6404 |
| Spotted gar | 183 | 668 | 91 | 646 | 124 | 480 | 74 | 363 | 146 | 1633 | 1787 | 6195 |
| Coelacanth | 62 | 3997 | 123 | 774 | 124 | 1463 | 74 | 265 | 146 | 1971 | 1763 | 10762 |

*exon 6 is located on a different contig from exons 1–5

#no RNA sequencing libraries express *Zmat2* mRNA

^overlaps exon of *Slbp2* gene

## Zmat2 loci in mammals and non-mammalian vertebrates

Fig 9 depicts maps of the *Zmat2* locus for a representative sampling of the vertebrates analyzed here, along with mouse *Zmat2* and human *ZMAT2*. In the 3 terrestrial species pictured [Unfortunately, this part of the genome in duck, flycatcher, Zebra finch, tropical clawed frog, and Chinese softshell turtle is poorly annotated, and thus is not shown], and in coelacanth, the locus exhibits several similarities in terms of the overall topology of at least 4 of the genes present, specifically *Ik*, *Hars*, *Hars2*, and *Zmat2*. Moreover, in spotted gar, even though the ortholog of *Hars2* is absent, the locus contains 3 other genes found in the terrestrial vertebrates, *Wdr55*, *Dnd1*, and members of the *Pcdh* family, although *Dnd1* is in a different relative location (Fig 9). Remarkably, these same 7 genes also are present in the human and mouse *ZMAT2*/*Zmat2* loci in a congruent orientation with terrestrial vertebrates, with the only difference being the relative transcriptional direction of *HARS2*/*Hars2* with respect to *Zmat2* (Fig 9).

**Table 7. Nucleotide identity with zebrafish *Zmat2* exons.**

| Species | Exon 1 (36 bp)* | Exon 2 (91 bp) | Exon 3 (124 bp) | Exon 4 (74 bp) | Exon 5 (146 bp) | Exon 6 s(643 bp)* |
|---|---|---|---|---|---|---|
| Tetraodon | no match | no match | 86.4 | no match | 88.2 | no match |
| Fugu | no match | 83.3 | 85.4 | no match | 88.2 | no match |
| Stickleback | no match | no match | 81.6 | no match | 82.6 | no match |
| Medaka | no match | no match | 87.9 | 85.7 | no match | no match |
| Cod | no match | no match | 86.7 | no match | 84.0 | no match |
| Tilapia | no match | no match | 83.5 | no match | 84.3 | no match |
| Amazon molly | 100 (18 bp) | no match | 82.5 | no match | 86.4 | no match |
| Platyfish | no match | no match | 83.5 | no match | 87.1 | no match |
| Spotted gar | no match | no match | 84.3 | 91.1 | 89.6 | 84.7 (98 bp) |
| Coelacanth | no match | no match | 81.5 | 89.2 | 89.3 | 94.7 (38 bp) |

*coding and non-coding DNA

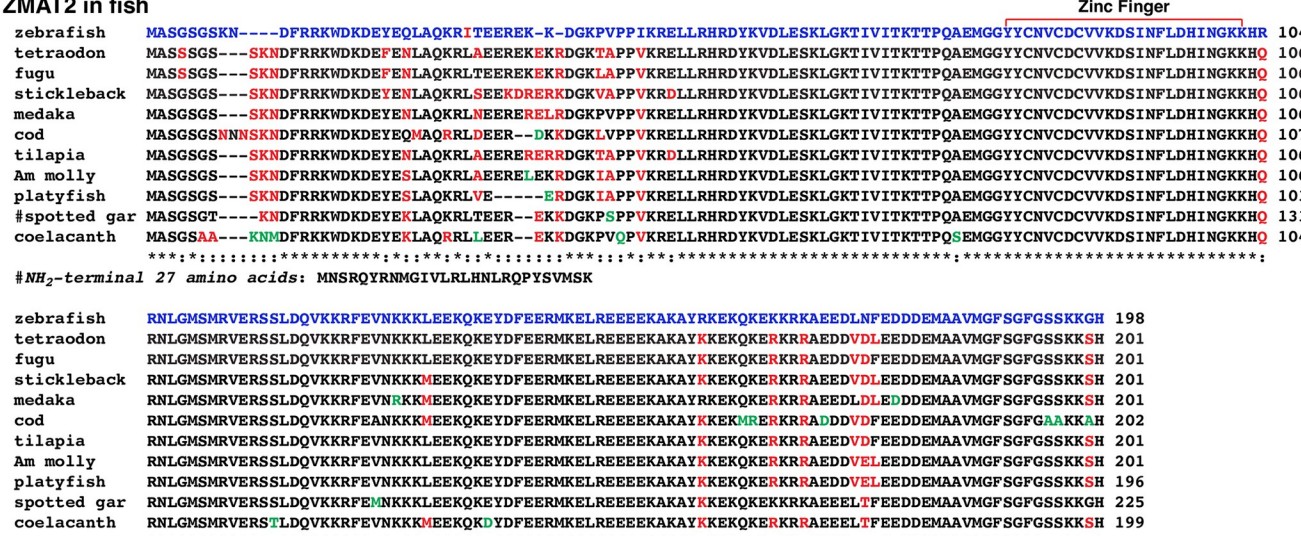

**Fig 8. Fish ZMAT2 proteins.** Alignments of amino acid sequences of ZMAT2 proteins from different fish are shown in single letter code. Identities and differences among proteins are indicated, with identities labeled by asterisks and differences by colons. Dashes indicating no residue have been placed to maximize alignments. Red and green text also depict differences, with amino acids marked in green indicating a unique change seen in just a single species. The zinc finger region has been labeled. As indicated by the #, spotted gar ZMAT2 has an additional 27 amino acids at its NH$_2$-terminus, as shown below. See Table 8 for additional information.

The *Zmat2* locus in teleost fish appears to be very different in the 5 species pictured here than in terrestrial vertebrates, coelacanth, spotted gar, or mammals (Fig 9). Except for *Zmat2*, none of the other genes found in the former species are present. In contrast, at least 2 additional genes are conserved among these teleosts, *Slbp2* and *Lcp2a* (*Lcp2* in Amazon molly).

## Discussion

The goal of this study was to characterize *Zmat2* genes in both terrestrial and aquatic vertebrates by analyzing data from genomic and gene expression resources. Prior to our recent publication [11], there had been no reports about *Zmat2* genes from any species, despite the

**Table 8. Identities with 198-amino acid zebrafish ZMAT2 in fish.**

| Species | Length | Percent Identity | Number of Differences |
|---|---|---|---|
| Tetraodon | 201 | 92.4 | 15 |
| Fugu | 201 | 93.4 | 13 |
| Stickleback | 201 | 89.9 | 20 |
| Medaka | 201 | 92.9 | 14 |
| Cod | 202 | 90.4 | 19 |
| Tilapia | 201 | 92.4 | 15 |
| Amazon molly | 201 | 92.9 | 14 |
| Platyfish | 196 | 92.9 | 14 |
| Spotted gar | 225 | 96.5 | 7 |
| Coelacanth | 199 | 91.9 | 16 |

*No NH$_2$-terminal methionine;

#NH$_2$-terminus undefined

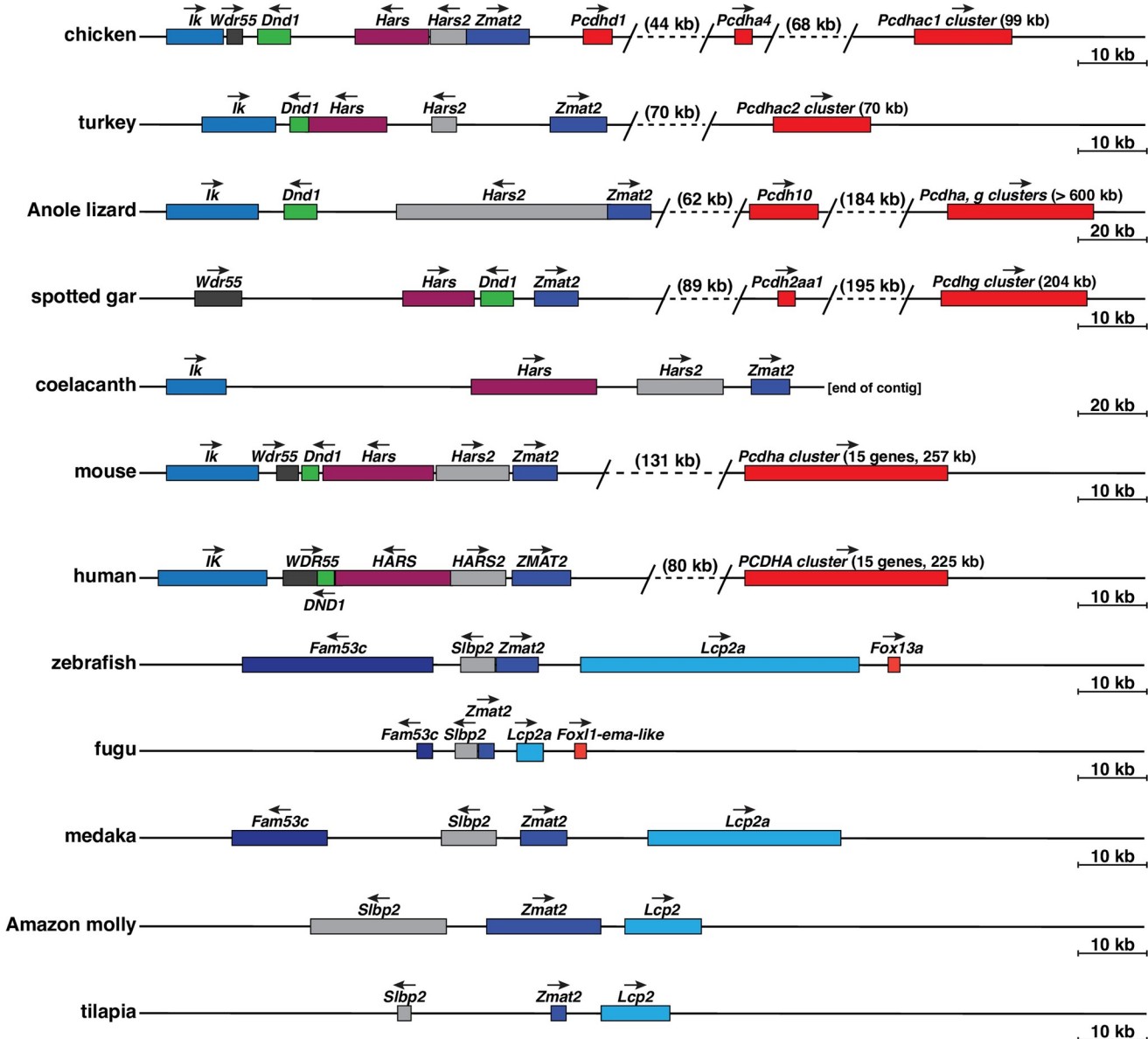

**Fig 9. The *Zmat2* locus in mammals and non-mammalian vertebrates.** Diagrams of the *ZMAT2/Zmat2* locus in chicken, turkey, Anole lizard, spotted gar, coelacanth, mouse, human, zebrafish, fugu, medaka, Amazon molly, and tilapia. The coordinates of individual genes are depicted as single boxes and color-coded, and include the following: *Ik/IK, Wdr55/WDR55, Dnd1/DND1, Hars/HARS, Hars2/HARS2, Zmat2/ZMAT2, Pcdh1, Pcdha4, Pcdhac1, Pcdha/PCDHA, Pcdhg* (gamma), *Pcdh10, Pcdh2aa1* (alpha a1), *Fam53c, Slbp2, Lcp2a, Lcp2, Fox13a, Foxl1-ema-like*. A horizontal arrow defines the direction of transcription for each gene. Scale bars are shown. Angled parallel lines indicate discontinuities, with the distances being spanned listed in parentheses.

importance of the protein in eukaryotic pre-RNA splicing, in which it is a component of the spliceosome [13,15]. The results here show that *Zmat2* is a single-copy gene in nearly all of the vertebrates that we have studied. This list includes teleost fish, in which their progenitors' entire genome had been duplicated ~320–350 Myr ago [22], and in which in some species such as zebrafish a significant fraction of duplicated genes has been retained [23]. As in mammals, *Zmat2* also is composed of 6 exons and 5 introns in most non-mammalian vertebrates (Figs 2 and 7), and parts of coding exons and of the protein are conserved (Figs 5 and 8, Tables

4 and 8), although not to the extent observed in mammals [11]. Additionally, even though *Zmat2* pseudogenes are present in half of the mammals that we have studied [11], they are rare in non-mammalian vertebrates. In fact, a putative pseudogene was identified in only one species, flycatcher, but as it is transcriptionally active (Fig 4D), unlike its mammalian orthologs, which are inert [24], it is actually a functional gene (here designated as flycatcher *Zmat2-2*; Fig 4). Collectively, these observations demonstrate that the focused analysis of a minimally studied gene such as *Zmat2* can lead to new insights about vertebrate biology and gene evolution, as well as provide an opportunity to develop new experimentally-testable hypotheses.

## *Zmat2* genes in terrestrial vertebrates

The data evaluated here show that *Zmat2* is a 6-exon gene in at least 5 of 8 terrestrial vertebrate species (Fig 2, Tables 2 and 3), and that its gene organization resembles that of its mammalian ortholog [11]. In flycatcher and Chinese softshell turtle, 2 of the 3 species with a different *Zmat2* gene structure, only 5 exons have been characterized (Fig 2), although it seems probable that the lack of identification of an exon 1 could be secondary to poor genome quality, especially since in flycatcher an equivalent of exon 1 is found in *Zmat2-2* (Fig 4). The other species with a *Zmat2* gene that was an outlier is *Xenopus tropicalis*. Its genome encodes an 8-exon *Zmat2* that contains 3 exons which do not correspond to those seen in other species (exons 5, 6, 7, Fig 3). As these exons were expressed in some *Zmat2* mRNAs (Fig 3), it is clear that alternative splicing of *Zmat2* occurs in frog tissues, and that these mRNAs most likely lead to different ZMAT2 proteins (Figs 4C–4E and 5B). The possibly distinct functions of these different proteins, which appear to be unique to the tropical clawed frog, are potential topics for future investigation into the biology of ZMAT2 in this species.

## A potential *Zmat2* pseudogene in flycatcher is expressed

Pseudogenes have been identified in prokaryotes and in eukaryotes [24], and appear to be relatively common in mammalian genomes, including in humans [24]. Analysis of data from ENCODE had reached the conclusion that over 10,000 pseudogenes are present in the human genome [25], and that nearly 80% of them are composed of processed mRNAs that had been retro-transposed as DNA back into the genome [25]. In our recent studies we identified between one to four *Zmat2* pseudogenes per genome half of the mammals analyzed, with most being DNA copies of processed and retro-transposed mRNAs that contained components of all 6 *Zmat2* exons [11]. None of these pseudogenes appeared to be transcribed in tissues in which authentic *Zmat2* was expressed [11]. However, since *Zmat2* pseudogenes were not detected in half of mammals examined, and since phylogenetic analysis showed greater similarity among pseudogene paralogs in a given species than to those in other mammals, we concluded that each had arisen independently following the evolutionary divergence of each mammalian species from its ancestors [11].

Remarkably, the only potential *Zmat2* pseudogene detected in a non-mammalian vertebrate was found in flycatcher, in which representations of all 6 exons were identified (Fig 4), even though no exon 1 had been mapped in 'authentic' flycatcher *Zmat2* (now *Zmat2-1*, see Figs 2 and 3). This putatively processed pseudogene was expressed, as has been demonstrated for a few other possible pseudogenes in other species [26], and thus *Zmat2-2* is an active gene in the flycatcher genome, although steady-state levels of its mRNA were several-fold less than *Zmat2-1* in brain, kidney, heart, liver, and testes (Fig 4D). However, taken together with observations about *Zmat2* pseudogenes in mammals, in which none of them appear to be transcriptionally active [11], it seems reasonable to postulate that in the flycatcher genome a future pseudogene has been identified in the midst of its evolutionary journey from active to inert.

## *Zmat2* gene and locus in fish

Results of analysis of *Zmat2* in 9 teleost and 2 non-teleost fish show that it is a 6-exon gene in all species studied (Fig 7, Tables 6 and 7), and thus appears to resemble mammalian *Zmat*2 [11]. However, the locus is different in teleost species than in mammals, terrestrial vertebrates, spotted gar, or coelacanth (Fig 9), in which *Zmat*2 maps adjacent to the *Hars2* and *Hars* genes (Fig 1A) [11]. In zebrafish, and in fugu, medaka, Amazon molly, and tilapia (Fig 9), *Zmat2* abuts *Lcpa and Slbp2*, with the latter being transcribed in the opposite direction than the former two genes (Fig 9). The genome was duplicated in a common progenitor of extant ray-finned fish ~320–350 Myr ago [22], and this duplication was followed by re-diploidization in ancestors of modern teleost lineages, with loss of one of the copies of the duplicated genes [22]. Based on the different location of *Zmat2* in these teleost species versus the non-teleosts, it can be postulated that the duplicated *Zmat2* gene was retained and the original lost during speciation.

## ZMAT2 proteins in non-mammalian vertebrates

In our recent analysis of ZMAT2 proteins in mammals, we found that they were remarkably similar to each other [11]. In fact, the 199-residue protein was identical in 11/18 mammals examined, and in the other species, there were just one or two amino acid substitutions, with most being conservative changes. Furthermore, no alterations were found in any mammal in the zinc finger domain [11].

In non-mammalian vertebrates, ZMAT2 is less conserved than in mammals. For example, among terrestrial vertebrates studied here, there are amino acid differences at 24 positions in the protein in addition to alterations within all of the $NH_2$-terminal 6 amino acids (Fig 5). Moreover, both flycatcher (ZMAT2-2) and *Xenopus tropicalis* ZMAT2 contain a single substitution in the zinc finger segment (Fig 5). In fish, the protein varies at 44 different locations among different species, including 26 within the $NH_2$-terminal region, although the zinc finger domain is invariant, and is identical with terrestrial vertebrates (except for flycatcher ZMAT2-2 and tropical clawed frog) and with mammals (compare Figs 4A and 8). Collectively, these results suggest that the zinc finger segment may be the most important region of ZMAT2, while perhaps the variable $NH_2$-terminal domain is the least important. Clearly, analysis of the functions of different domains of the protein by biochemical, genetic, or other approaches will be an important area for future investigation.

## Improving the quality of individual genes in genome databases

Genomic databases are valuable resources for the entire scientific community as they contain information that can be used to develop and sharpen evolutionary and physiological relationships about a large number of species, and can serve as the springboard for new discoveries. As noted here, the quality of data on *Zmat2* in Ensembl is poor. In only 6 of nineteen species examined, the annotated *Zmat2* gene included both 5' and 3 UTRs (Tables 1 and 5), and in some species postulated *Zmat2* exons were not detected in *Zmat2* mRNAs. These types of problems appear to be common, as they were found for mammalian *Zmat2* genes in the same genome databases [11], and have been identified for other genes too [27,28]. It clearly will be very important to improve overall data quality in these resources for future investigations to begin accurately.

## Final conclusions

Remarkably, as noted recently, ~90% of human genes are understudied [7], and it seems likely that this is true for an even greater fraction of genes in non-mammalian vertebrates. The opportunity to evaluate and examine in detail the large number of unstudied and understudied genes and gene families has the potential to lead to new discoveries of both fundamental and biomedical importance. In fact, *Zmat2* exemplifies this opportunity because of the defined role of its protein product in pre-mRNA splicing [13,15]. The recent discovery that splicing factors can feed back and regulate transcription of alternative promoters for protein-coding genes [29,30] demonstrates how an integrated understanding of the interplay between the spliceosome and chromatin-modifying enzymes to control RNA polymerase II activity can lead to new insights with evolutionary and practical implications.

## Supporting information

**S1 Fig.**
(PDF)

**S1 Table. RNA-sequencing libraries screened for gene expression.**
(DOCX)

**S2 Table. Probes for screening RNA-sequencing libraries.**
(DOCX)

## Author Contributions

**Conceptualization:** Peter Rotwein.

**Formal analysis:** Peter Rotwein.

**Funding acquisition:** Peter Rotwein.

**Investigation:** Peter Rotwein.

**Methodology:** Peter Rotwein.

**Project administration:** Peter Rotwein.

**Writing – original draft:** Peter Rotwein.

**Writing – review & editing:** Peter Rotwein.

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
