## [Decision Letter · Decision Letter 0]

29 Apr 2020

The Zmat2 Gene in Non-mammalian Vertebrates: Organizational Simplicity within a Divergent Locus in Fish

PONE-D-20-01473

Dear Dr. Rotwein,

We are pleased to inform you that your manuscript has been judged scientifically suitable for publication and will be formally accepted for publication once it complies with all outstanding technical requirements.

With kind regards,

Livia D'Angelo

Academic Editor

PLOS ONE

Reviewers' comments:

Reviewer's Responses to Questions

**Comments to the Author**

1. Is the manuscript technically sound, and do the data support the conclusions?

Reviewer #1: Yes

Reviewer #2: Yes

2. Has the statistical analysis been performed appropriately and rigorously? 

Reviewer #1: Yes

Reviewer #2: N/A

3. Have the authors made all data underlying the findings in their manuscript fully available?

Reviewer #1: Yes

Reviewer #2: Yes

4. Is the manuscript presented in an intelligible fashion and written in standard English?

Reviewer #1: Yes

Reviewer #2: Yes

5. Review Comments to the Author

Reviewer #1: In this paper, the Author focuses on the Zmat2 gene in lower (non-mammalian) vertebrates. This paper extends previous work in mammals by the same author's group. By using a combination of database/dataset analysis and in silico search in gene expression repositories, the author completes the gene and protein sequence framework in vertebrates, thus offering a large and comparative picture of differences and similarities along the vertebrate scale. Such comparative approach may be useful in further studies including biomedical and translational ones.

The paper is easy to read and straightforward. Defining Zmat-2 in vertebrates represents an example of how in silico analyses may easily contribute to basic research around underestimated genes.

Major points

None

Minor points

See e.g. page 22, line 480 'halfof'

Reviewer #2: In “The Zmat2 Gene in Non-mammalian Vertebrates: Organizational Simplicity within a

Divergent Locus in Fish”, Peter Rotwein describes the evolutionary history of the zinc-finger Zmat2 gene and protein in non-mammalian vertebrates, following his recent twin paper on the characterization of the mammalian orthologs (doi.org/10.1186/s12864-020-6506-3). These types of studies are highly relevant to understanding the genomic mechanisms underlying evolution of key genes (such as spliceosome-related factors), to exploit genomic and gene expression databases (highlighting database-specific and species-related pros and cons) and to gain and develop functional insights and hypotheses. The author mines a wide array of vertebrate loci to determine degrees of conservation and divergence of Zmat2 gene/protein structure/sequence in order to reconstruct the phylogenetic history. This paper describes interesting issues concerning the nature and dynamics of gene evolution, including pseudogenes and copy number variation, and to me represents an important effort with high potential to tackle down the molecular mechanisms of spliceosome functions. The overall text is well structured and very accurate, with no fundamental flaws and weaknesses, and so will be accessible to the general readership of PLOS One. These results may provide cues that are important to scientists interested in spliceosome and keratinocyte research. I consider this manuscript a very suitable original contribution. I do not have any major criticisms to suggest in any regards. Actually, I found only a couple of typos, one being ‘and’ in italics in line 534. Therefore, I strongly support publication of this manuscript.

6. PLOS authors have the option to publish the peer review history of their article (what does this mean?). If published, this will include your full peer review and any attached files.

Reviewer #1: No

Reviewer #2: No

---

## [Editor Report · Acceptance letter]

7 May 2020

PONE-D-20-01473 

The *Zmat2* Gene in Non-mammalian Vertebrates: Organizational Simplicity within a Divergent Locus in Fish 

Dear Dr. Rotwein:

I am pleased to inform you that your manuscript has been deemed suitable for publication in PLOS ONE. Congratulations! Your manuscript is now with our production department. 

With kind regards,

on behalf of

Dr. Livia D'Angelo 

Academic Editor

PLOS ONE